# Negative Design Margin Realization through Deep Path Activity Detection Combined with Dynamic Voltage Scaling in a 55 nm Near-Threshold 32-Bit Microcontroller

**DOI:** 10.3390/s23177498

**Published:** 2023-08-29

**Authors:** Run-Ze Yu, Zhen-Hao Li, Xi Deng, Zheng-Lin Liu

**Affiliations:** School of Integrated Circuits, Huazhong University of Science and Technology, Wuhan 430074, China; d201980708@hust.edu.cn (R.-Z.Y.); zhenhao@hust.edu.cn (Z.-H.L.); d202180903@hust.edu.cn (X.D.)

**Keywords:** dynamic voltage scaling (DVS), negative design margin, error detection and correction (EDaC), ultra-low voltage, near-threshold operation, high stability, error-aware capability

## Abstract

This paper presents an innovative approach for predicting timing errors tailored to near-/sub-threshold operations, addressing the energy-efficient requirements of digital circuits in applications, such as IoT devices and wearables. The method involves assessing deep path activity within an adjustable window prior to the root clock’s rising edge. By dynamically adapting the prediction window and supply voltage based on error detection outcomes, the approach effectively mitigates false predictions—an essential concern in low-voltage prediction techniques. The efficacy of this strategy is demonstrated through its implementation in a near-/sub-threshold 32-bit microprocessor system. The approach incurs only a modest 6.84% area overhead attributed to well-engineered lightweight design methodologies. Furthermore, with the integration of clock gating, the system functions seamlessly across a voltage range of 0.4 V–1.2 V (5–100 MHz), effectively catering to adaptive energy efficiency. Empirical results highlight the potential of the proposed strategy, achieving a significant 46.95% energy reduction at the Minimum Energy Point (MEP, 15 MHz) compared to signoff margins. Additionally, a 19.75% energy decrease is observed compared to the zero-margin operation, demonstrating successful realization of negative margins.

## 1. Introduction

The demand for energy-efficient digital circuits has surged, driven by the burgeoning applications such as Internet of Things (IoT) devices and wearables, prompting the emergence of near-threshold computing (NTC) [1,2,3]. Despite its merits, NTC brings about considerable degradation in path delay due to process, voltage, and temperature variations, as well as aging effects. The operational point of a chip with minimal stability margin is referred to as the zero-margin point. Traditional chip signoff methods, when applied under worst-case scenarios, result in a significant margin increment for NTC chips, hindering the realization of projected power reduction advantages [1,2,3]. As such, achieving reduced system energy consumption by compressing margins through voltage reduction is contingent upon maintaining chip operational speed.

The simplest recourse for margin alleviation entails the use of a replica delay line for on-chip performance monitoring [4]. Nevertheless, replica systems are inadequate in negating margins for local variations, such as intra-die discrepancies, local resistive (IR) drops, and localized temperature hotspots that elude capture.

To surmount margin concerns across various forms of variations, a multitude of error detection and correction (EDaC) strategies have been postulated [5,6,7,8,9,10,11,12,13,14,15,16,17,18,19,20]. These approaches pivot around detecting data transitions within a predefined window following the rising edge of the clock signal. A match within this window denotes a timing error occurrence, necessitating the application of architectural-level measures to correct the identified error. Differences in the specifics of implementation may arise due to factors such as the nature of timing elements (flip-flops or latches), the method of window generation, the technique for data transition detection (current-based or voltage-based), and the strategy for timing error correction (pipeline flushing, clock borrowing, instruction replay, etc.). Two predominant issues intertwine with EDaC methodologies: firstly, the tension between detection window (DW) width—indicative of error-aware capacity—and the subsequent addition of hold buffers, leading to area overhead; secondly, the substantial performance drop stemming from frequent timing error corrections, obstructing the attainment of a zero-margin state.

In response to these prevailing challenges, this study offers a novel rendition called Deep Path Activity Detection–EDaC (DPAD–EDaC). This novel approach integrates a timing error prediction circuit into the conventional EDaC framework, adeptly addressing pivotal concerns and achieving negative margin operation at a controlled error rate. A programmable width-prediction window is established just prior to the root clock’s rising edge. Detection cells, interleaved with combination logic units in the timing path, emit pulse signals upon discerning data alterations. The presence of a pulse signal within the prediction window signifies a potential timing error occurrence. Since the timing error remains latent when predicted, rectification becomes feasible through a simplistic one-clock cycle enablement mechanism. Thus, the system can function at a pre-determined predicted error rate, characterized by marginal performance loss. This operational schema, operational at diminished power supply voltages, ultimately realizes a design replete with negative margins.

In practical chip operations, the precision of timing error prediction is significantly impacted by variations in clock network delay and data path delay, exacerbated in near-threshold conditions. To counteract these effects, a dynamic voltage regulation circuitry is introduced, calibrating the system’s timing error prediction capability and power supply voltage according to EDaC circuitry’s error detection outcomes. This adaptive calibration ensures stable system operation at the specified error rate, accommodating clock network and data path delay influences, particularly in near-threshold contexts.

This dual-pronged strategy, amalgamating error prediction enhancement and dynamic voltage regulation, epitomizes our proposed approach, culminating in the achievement of stable operations with negative margins. Empirical evaluation and experimental validation are undertaken via the deployment of a near-threshold 32-bit ARM Cortex M0 microprocessor system. The ensuing outcomes underscore the effectiveness of our methodology in realizing negative design margins while concurrently optimizing system performance and energy efficiency. The proposed approach is realized through a 55 nm CMOS implementation within a near-threshold 32-bit ARM Cortex M0 microprocessor system, demonstrating a modest 6.84% area overhead compared to the baseline non-EDaC system. Experimental findings corroborate a 46.95% energy reduction at the Minimum Energy Point (MEP, 15 MHz) when contrasted against signoff margins, and a 19.75% energy decrease relative to the zero-margin operation, underscoring the achievement of negative margins.

## 2. Overview of EDaC Techniques

### 2.1. Error Detection

In most EDaC systems, timing error detection is implemented by double sampling (DS). It relies on the fact that the sample at T2 is more likely to be correct than the sample at T1 (T2 > T1). Several works [7,8,9] implement DS by adding a shadow latch to a sequential element. The rising and falling edge of the clock are used as T1 for the sequential element and T2 for the shadow latch. If both samples do not match, this is flagged as a timing error. In order to alleviate area overhead and constraints on the clock, DS is gradually replaced by transition detection (TD). A late activity in DW detected by the TD cell is flagged as a timing error. Several works [11,12,13,14,15,16,17,21,22,23,24] insert a TD cell at the data port of the timing element or integrate it into the element. It should be noted that the error-aware capability of the DS-/TD-based system is limited to timing errors on FF- or latch-based endpoints. This leaves critical paths toward input–output (IO) cells and macros (e.g., SRAM) unprotected. Generally, this limitation is overcome by splitting unprotected critical paths into several non-critical paths, which adds to the design complexity.

### 2.2. Detection Window

In DS or TD EDaC systems, DW provides a timing range for error detection. However, this brings additional hold constraints for all monitored endpoints to avoid false errors due to short-paths’ conduction. Some works [9,12,13,14,21,24] reuse the system clock as DW, and DW width adjustment is achieved by clock duty tuning. This implementation imposes a large number of hold buffer insertions and continues to grow as supply decreases. Several works [10,11,17] add a dedicated DW circuit to balance the conflict between area overhead and error-aware capability, which is difficult to quantify.

### 2.3. Error Correction

All error correction strategies can be divided into the following three categories: (1) correction, (2) prediction, (3) masking. Strategy (1) is usually used in DS- or TD-based EDaC systems to ensure operation near the point of first failure (PoFF) with little margin. In [11,24], the corrections were achieved through instruction replay or bubble insertion at the cost of significant cycle overhead, wmaking them difficult to apply to different processors. Both (2) and (3) are preferred in systems operating beyond the PoFF to achieve more energy savings with little overall loss owing to clock gating/stretching. In (2), timing errors are predicted and eliminated before the clock rising edge. In [25,26], the completion of logic operations in the datapath was performed at the end of the clock period to determine, instantaneously, the presence of late-arriving signals. This strategy omits the correction circuit but brings out the number of false errors due to mismatch between the PW and the clock of endpoints caused by clock latency. Finally, in (3), late arrivals are passed to the next cycle, which allows systems to operate without disturbance due to timing errors. Usually, this approach is combined with a latch-based pipeline, which is achieved using the clock-borrowing feature of the latch. In [13,19,20], this approach was combined with a latch-based pipeline to enable time borrowing. However, the maximum borrowing cascade limit needed to be imposed to avoid the risk of system failures. Meanwhile, it is difficult to eliminate the adverse impact of glitches on latch-based systems [27,28].

## 3. Presented Concept and Analysis

To address the shortcomings in the existing EDaC strategies, this work proposes the deep path activity detection–EDaC (DPAD–EDaC) strategy, consisting of lightweight error prediction, detection, and correction circuits. The benefits of error prediction before the clock edge (e.g., simple correction and strong error-aware capability) are retained. And the critical problems (e.g., false prediction errors in EPC and critical tradeoff in EDC) are addressed. Finally, the DPAD–EDaC system operates at a certain error rate with the support of DVS to achieve negative design margin.

### 3.1. Error Prediction Circuit (EPC) Concept

As shown in Figure 1, timing error prediction relies on the following two operations. First, a TD cell (yellow shadow) consisting of a delay chain, and an XOR-gate flags toggle activity in the critical path by outputting a high signal with pulse width $T_{H}$. Next, the DYN-OR TREE (blue shadow) evaluates the outputs of all TDs at the end of the clock cycle (PW) and reduces them to a single prediction error signal (Fp_error). The generation process of the Fp_error signal is shown in the timing diagram, shaded in purple. It is obvious that PW is limited to a time range exactly before the rising edge of the root clock (clk_root) due to the clock gating/stretching method for error prevention.

In an ideal clock network, clk_root coincides with the clock of a sequential endpoint ($clk\_dff_{des}$). A high Fp_error signal indicates that ongoing critical activity in PW is close to the edge of $clk\_dff_{des}$, which is equivalent to an actual timing error. However, there is a latency ($T_{latency,des}$) between PW and the edge of $clk\_dff_{des}$ due to the clock network latency in an actual chip, which increases rapidly with voltage decrease. The resulting inequivalence between high Fp_error signal and timing error brings a plethora of false errors for the EP systems in works [25,26]. To address this problem, we first push the monitored cells deeper into critical paths to precisely compensate for the clock latency and re-establish the equivalence. However, the variable datapath propagation delay and clock latency will continue to destroy the equivalence, increasing the difficulty of monitored cell selection. Next, the PW width is no longer fixed, but is instead precisely adjusted by the results of EDC. The process of adjustment is also considered as the equivalence re-establishment process. Meanwhile, ECC is necessary to ensure correct system operations during the adjustment process.

In this paper, PW is generated by the PW_GEN module (top left in Figure 1) and adjusted by PW_status (∈[0,7] in this paper) signal from the dynamic window scaling (DWS) module. As stated above, the DYN-OR TREE executes the dynamic behavior, which takes care of the instant sampling of TDs’ outputs during PW. To guarantee reliable sampling, two constraints for PW width are proposed:As the supply decreases after power-up, timing errors are gradually detected by EDC, and PW_status increases accordingly until one of the TDs’ outputs is covered. Therefore, the difference between adjacent PW width should be greater than a TD’s output pulse width $T_{H}$, which needs to be satisfied at different PVT corners:
(1)$${\forall i\in \left[0,6\right]:\;PW}_{PW\_status=i+1}\;-\;{PW}_{PW\_status=i}>T_{H};$$A minimum pulse width constraint applies on the $PW_{PW\_status=1}$. This constraint equals the sum of the worst case propagation delay for DYN-OR TREE and setup time for Integrated Clock Gating (ICG) cell:
(2)$${PW}_{PW\_status=1(ff)}>T_{prog,OR(ss)}\;+\;T_{setup\left(ss\right)}.$$

It is clear that the interval between ${PW}_{PW\_status=1}$ and ${PW}_{PW\_status=7}$ is the safe propagation range (SRR) for each TD’s output. This means that TDs’ outputs covered by SRR can be accurately evaluated and propagated through DYN-OR TREE during PW adjustment, that is, these cells can be monitored.

As shown in Figure 2, the path is placed on a timeline with the rising edge of clk_root as the origin. For each path, $T_{arrival}$ (arrival time for path), $T_{need}$ (latest arrival time limit), and $T_{PW\_status=i,left}$ (the left boundary of PW) can be expressed by constants (i.e., $T_{latency,source}$, $T_{datapath,prog}$, $T_{latency,des}$, $T_{setup}$ and ${PW}_{PW\_status=i}$) at defined PVT conditions: (3)$$\begin{array}{c} T_{arrival}=T_{latency,source}+\;T_{datapath,prog}T_{need}=T_{period}+\;T_{latency,des}-\;T_{setup}\\ T_{PW\_status=i,left}=T_{period}-{PW}_{PW\_status=i.} \end{array}$$

Firstly, the $T_{period}$ when timing error occurs coincidentally can be inferred by letting $T_{arrival}=T_{need}$; then, the $T_{PW\_status=i,left}$ in the timeline can be further obtained for each path: (4)$$T_{PW\_status=i,left}=T_{datapath,prog}+T_{latency,source}-T_{latency,des}+T_{setup}-{PW}_{PW\_status=i.}$$

Next, the SRR and the left boundary of PW at each PW_status can be determined by changing the value of i. Finally, monitored cells can be further selected. In this work, we select the cell just covered by $T_{PW\_status=5,left}$ (the middle of SRR) as the FMC (First Monitored Cell) for each critical data path. Then, additional monitored cells are selected with a maximum interval constraint to maintain continuous error-aware capability under severe variations. This constraint applies to the adjacent monitored cells, which equals the minimum interval between $T_{PW\_status=5,left}$ and $T_{PW\_status=1,left}$: (5)$$\begin{array}{c} T_{{TD}_{ci},ss}-\;T_{{TD}_{ci+1},ss}>{PW}_{PW\_status=5,left,ff}-\;{PW}_{PW\_status=1,left,ff.} \end{array}$$

This ensures that there are always monitored cells within SRR to maintain continuous error-aware capability. It follows that this capability is guaranteed by discontinuous monitored cells.

### 3.2. Error Detection Circuit (EDC) Concept

As suggested above, the PW width is precisely adjusted by the results of EDC, which relies on the following three operations, as shown in Figure 1. First, a TD cell flags toggle activity of the critical endpoint’s data input by outputting a pulse signal. Next, all TDs are grouped and the TD outputs in each group are evaluated and latched by the shared error latch (SEL in Figure 3) in Error Latch Line (gray shallow in Figure 1) within the DW. The DW generation (DW_GEN) cell (green shadow in Figure 1), consisting of a delay chain and a NAND2 gate, flags a timing range for error detection by outputting a high signal with pulse width $T_{DW}$ at the start of the clock. Finally, the Error-OR Tree (orange shadow in Figure 1) ORs all outputs to a single detection error signal (Fd_error). The generation process of the Fd_error signal is shown in the timing diagram shaded in orange in Figure 1.

In more detail, the number of TD cells in each group and the fan-in of the corresponding SEL cell should be equal. We can also configure it flexibly according to design requirements. The design of multiple endpoints sharing one SEL cell can effectively reduce the area overhead of EDC.

As described in Section 2.2, the conflict between DW width and area overhead brings great difficulties to conventional EDaC systems. In this paper, this conflict is naturally attenuated. The results of EDC help with the adjustment of PW and, more specifically, the re-establishment of the equivalence between prediction result and actual timing error. Thus, the system’s accurate operation is guaranteed by prediction, which weakens the constraint on DW width imposed by error-aware capability. Despite that, the process of voltage decreasing from standard voltage (1.2 V) to the appropriate low voltage needs to be specially considered. During the process, the DW must be wide enough to detect all errors caused by any voltage decrease step. As shown in Equation (Equation 6), the minimum DW width (FF corner) needs to be greater than the maximum delay difference (SS corner) between the critical paths at any two adjacent voltages: (6)$${\forall i\in \left[0,31\right]:\;DW}_{pulse,ff,vol\_status=i}>T_{cpath,ss,vol\_status=i}-\;T_{cpath,ss,vol\_status=i+1}\;$$

In addition, considering the meta-stability of the timing elements [29,30], a TD’s output pulse width $T_{H}$ should be greater than the sum of the setup time for the critical endpoint and hold time for the connected SEL: (7)$$T_{H}>T_{setup,critical\;endpoint}+\;T_{hold,SEL.}\;$$

In this way, the meta-stability event can be detected.

### 3.3. Error Correction Circuit (ECC) Concept

The error correction operation is necessary when a timing error occurs. However, the correction methods proposed in works [11,24] are strongly correlated with the processor architecture, which imposes limitations on the implementation in other systems [31]. To address this problem, we propose an instruction replay method based on ARs’ backup for error correction. The concrete implementation of ECC is shown in Figure 4, where the changes in ARs are backed up at each instruction boundary. In this work, each instruction modifies up to four ARs (PC, PSR, C bit, and one of R0–R15) at the same time, which is an inherent characteristic of the ARM Thumb-2 ISA. Thus, the backup block consists of four memory rows with a low area overhead. A high Fd_error signal flushes and stalls the global pipeline until completing voltage or PW regulation with a high replay signal. Then, the modified ARs are overwritten with the corresponding value in the backup block, and the processor returns to normal operation.

The validity of this correction technique can be generalized to more processors by distinguishing instruction boundaries, the maximum number, and categories of modified ARs for each instruction.

### 3.4. DVS and DWS Module

As shown in Figure 5, the integration of the DPAD–EDaC strategy can be abstracted as the implementation of a distributed error processor (DEP), consisting of distributed EPC, EDC, and the error rate comparison circuit. The signals from DEP are involved in DVS and DWS modules to ensure correct system operation at a certain prediction error rate for negative margins.

The error rate comparison circuit consists of a timer, counter, and comparator, which are driven by the ungated clock (clk_del) to calculate the prediction error rate and compare it with high and low thresholds. The registers in the timer (counter) start self-incrementing (counting Fp_error signal) from the configured initial value until the overflow signal OV_T (OV_C) or reset signal OV_C (OV_T) are generated. In more detail, the high OV_T signal represents timing completion and the high OV_C signal represents the error rate exceeding the high threshold. Meanwhile, the high OV_T signal enables comparator to compare the counting result with the low threshold, and a high MER signal is generated for voltage decrease if it is lower.

The high Fd_error signal from EDC controls the DWS module to increase PW_status and generate the high replay signal for instruction replay. However, if PW_status is about to overflow or the OV_C signal is high, the DVS module is controlled to increase supply voltage. Then, the high replay signal is generated when receiving the completion signal from LDO.

Considering the situation of voltage regulation loop with impact of temporary variations (i.e., temperature variations, IR drop), a voltage–PW_status pair dictionary is added to reduce the resulting cycle overhead. When the voltage is adjusted from level A to B, the PW_status will be stored in the dictionary with index A and read out with index B, and then PW width will be adjusted correctly within a cycle.

## 4. Implementation

As shown in Figure 6, the DPAD–EDaC strategy was implemented in a near-threshold 32-bit microprocessor system. It consisted of a CORTEX-M0 core, 2KB self-designed stable 7T SRAM, distributed error processor, DVS module, DWS module, and several peripherals.

### 4.1. Ultra-Low Voltage Implementation

This implementation targets 5–100 MHz operation speed, with voltage swept from 0.4 to 1.2 V in steps of 25 mV. To facilitate the ultra-low voltage implementation, the operation of all standard cells is verified at the concerned voltage. Cells with functional errors or extremely large delays are excluded, and retained cells are recharacterized at these voltages to obtain new timing libraries. Next, a low-power synthesis and place and route (P&R) flow translates the microprocessor register transfer level (RTL) to silicon. During this process, the system is split into three power domains, as shown in Figure 6. It is noteworthy that SRAM is placed in the regulated low-voltage domain for lower energy consumption. Considering the read failure problem of conventional 6T-SRAM at low voltages [32,33,34], a customized stable 7T-SRAM with error detection capability suited to EDaC systems was used.

The integration of DPAD–EDaC can also be interpreted as the process of transforming design to integration for EPC and EDC. Further, the design flow was expanded with a series of automated steps, as shown in Figure 7. Firstly, static statistical timing analysis was performed after the initial P&R to determine the design details of EDC and EPC.

### 4.2. EDC Design Details

Monitored cell selection is one of the main focuses in EDC design. The maximum clock frequency is determined by critical paths. Thus, only the most critical endpoints need to be monitored, allowing a limited overhead. Figure 8 shows a histogram of all the endpoints, ordered with respect to the smallest timing slack path they serve. The number of monitored endpoints is determined by the chance of false positive monitoring of the EDC, e.g., the chance that a non-monitored path propagates more slowly than all monitored paths owing to various variations, as described in Equation (Equation 8): (8)$$P_{EDC,false}=P(\exists \;p_{i}:T_{prop,p_{i}}>\max\left(T_{prop,q_{j}}\right)$$
where $p_{i}$ represents all non-monitored paths and $q_{i}$ represents all monitored paths. When false monitoring occurs, it is probable that non-monitored paths have failed but monitored paths have not, thus causing a system operation failure. To avoid this occurrence, enough timing slack (12.2% Tperiod in this paper) is covered, with 343 out of 4025 endpoints being monitored. The probability of such an event was determined by the delay distribution of a subset of paths from 1000 Monte Carlo (MC) simulations at 500 mV (see Figure 9), which was less than 1 × $10^{-15}$ in this paper, decreasing with increased voltage.

Then, the TD cells are inserted at the data points of all monitored endpoints. Equation (Equation 7) constrains the minimum pulse width of the TD cell. The pulse width $T_{H}$ is determined by its internal delay chain. An initial TD implementation with existing standard cell results in an area footprint of 10.08 µm^2^. To minimize area, stacked transistors with long gate-length due to RSCE (Reverse Short-Channel Effect) are used in the delay chain. This custom TD cell reduces the area footprint to 6.72 µm^2^.

Another focus of EDC design is the DW_GEN (DW generation) cell design. The EDC is equipped with a 7% Tperiod DW width satisfying the constraint in Equation (Equation 6). Then, the DW_GEN cell can be implemented and optimized with the same internal delay optimization methods as in a TD cell. Meanwhile, this results in 608 hold time buffers as overhead, as shown by the selected point in Figure 10. Compared to conventional EDaC systems with 10% Tperiod DW (1789 hold buffers), a 68.59% reduction in hold buffers is achieved. This owes to the weakening correlation between DW width and error aware capability in the DPAD–EDaC system.

### 4.3. EPC Design Details

One of the main focuses in EPC design is the PW_GEN (PW generation) cell design. As suggested above, Equation (Equation 2) constrains the minimum width of $PW_{PW\_status=1}$, and Equation (Equation 1) constrains the minimum PW width difference between adjacent PW_status. Then, the PW_GEN cell can be implemented and optimized similarly using the DW_GEN cell.

Another focus in EPC design is the monitored cell selection. Cells can be selected from the static timing reports, which directly influence the performance of the prediction system. For each path towards monitored endpoints, the defined values obtained from the timing report are substituted into Equations (Equation 4) and (Equation 5). Then, all monitored cells can be selected discontinuously. We keep in mind that the DVS loop is able to re-scale overall circuit performance to a pre-defined error rate. The system does not require excessive error prediction capability. In total, when 125 monitored cells are selected, the sparse selection in this paper achieves 87.5% area reduction compared to the 1000 monitored cells selected in works [25,26].

Next, all TDs are connected to the monitored cells and endpoints for integration. To minimize extra wire loading on the existing paths, the TDs are placed as close as possible to the selected cells. All layout modifications are made using engineering change of order (ECO) commands to avoid large changes to the existing layout. In more detail, 343 TDs in the EDC are divided into 57 groups based on the position of endpoints in the clock network. Each group is connected to an SEL cell that is placed centrally in the group, which helps to minimize the overall routing overhead. The same procedure is repeated to obtain other connections in EDC. Then, 125 TDs in the EPC are divided into groups of 5 by a k-means clustering algorithm based on their location, and the centrally placed procedure is repeated for other connections in EPC. All new placement and routing are carried out using ECO commands, and a new retiming is verified with all constraints. Further iterations of clustering/placement are performed until timing is closed. In Figure 11, the DPAD–EDaC system is implemented in 55 nm CMOS process and the die photograph is shown. Here, the EDaC insertion infers a 6.84% area overhead and the details are shown in Figure 12.

## 5. Experimental Results

This section reports the voltage and energy experimental results at different operating conditions by the post-simulation and the power analyses.

### 5.1. Experimental Setup

The system was established to be fully functional by running a Dhrystone benchmark in a frequency range from 5 to 100 MHz under four voltage scaling conditions: (1) zero-margin critical voltage scaling ($V_{critical}$); (2) signoff voltage scaling ($V_{signoff}$); (3) PoFF voltage scaling ($V_{PoFF}$); (4) 5% error rate voltage scaling ($V_{DPAD-5\%ER}$).

In the first condition ($V_{critical}$), the system operates with a zero margin at the most critical point. This can be obtained from the static timing analysis (STA) results by lowering the voltage until the slack of the most critical path equals zero at TT corner and 20 ${}^{{}^{\circ}}$C. In the second condition ($V_{signoff}$), the system operates with a conventional signoff margin. The margin is obtained from the same procedure as $V_{critical}$, except that the condition is changed to SS corner, 125 ${}^{{}^{\circ}}$C, and 10% supply drop. In the third condition ($V_{PoFF}$), the system operates more critically than the signoff margin. Here, the supply voltage is adjusted according to error detection results from EDC, with EPC closed. The PoFF is determined as 1 detection error per 10,000 cycles at TT corner and 20 ${}^{{}^{\circ}}$C. In the fourth condition ($V_{DPAD-5\%ER}$), the system operates the most critically, with negative margins at a 5% error rate tolerance point. Here, the supply voltage is adjusted by the DVS module according to the error rate at TT corner and 20 ${}^{{}^{\circ}}$C.

### 5.2. Operation Comparison under Four Conditions

Figure 13 compares the voltage scaling of $V_{critical}$, $V_{signoff}$, $V_{PoFF}$, and $V_{DPAD-5\%ER}$ over the frequency range. The area between $V_{DPAD-5\%ER}$ and $V_{signoff}$ (pink shadow) represents the voltage margin reduction compared to a conventional baseline design, which equals 195 mV at the MEP (15 MHz). And the area between $V_{DPAD-5\%ER}$ and $V_{c}ritical$ (slope shadow) represents the voltage margin reduction compared to a zero-margin design, which equals 10 mV at the MEP. Looking at the $V_{DPAD-5\%ER}$ curve shows that the DPAD–EDaC system demonstrates less voltage margin reduction at near-/sub-threshold voltage. This trend is caused by an over 5% error rate increase due to propagation delay deterioration of critical paths when the voltage status decreases (25 mV drop) in the near-/sub-threshold region.

Next, Figure 14 shows the energy consumption analysis under these voltage scaling conditions. The area between $V_{DPAD-5\%ER}$ and $V_{signoff}$ (purple shadow), $V_{DPAD-5\%ER}$, and $V_{critical}$ (diagonal shadow) represent the energy savings compared to a conventional baseline design and a zero-margin design, which equal 14.02 pJ/cycle and 3.74 pJ/cycle at the MEP, respectively.

### 5.3. Comparison

A more detailed comparison between the proposed DAPD-EDaC and existing near-threshold capable EDaC-based systems is provided in Table 1. The top part of the table focuses on strategy integration details (e.g., DW width, area overhead, and low-voltage stability). This shows that our work has a significantly high error-aware capability with low DW width and minimal area overhead. At the same time, the widespread failures existed in most prediction systems due to large clock latency at low supply voltage are eliminated in this work.

Next, the middle part of the table shows the process node, architecture, voltage and frequency range for functional operation. This indicates that the proposed design is able to operate within a large voltage and frequency range that goes from nominal to subthreshold region.

Finally, the bottom part shows the achieved energy savings. Setting the predicted error rate at 5%, we subjected the chip to Dhrystone benchmark test programs for subsequent post-simulation validation. The simulation outcomes indicate that, at the MEP (15 MHz), the chip can reliably operate at 0.45 V. In comparison, this voltage value is 0.67 V for the conventional baseline (non-EDaC) design and 0.48 V for critical operations. Moreover, this research attains a 47.87% reduction in energy consumption compared to the conventional baseline (non-EDaC) design and a 19.75% energy reduction in comparison to the zero-margin critical operation. Benefitting from the real-time adjustments made by the DVS module to power supply voltage and the prediction window, our design can lower the power supply voltage below the zero-margin point with a commensurate performance loss, as per the preset error rate. Notably, this margin is present in other works, remaining at a minimum of 12%.

## 6. Conclusions

In conclusion, our research introduces an innovative approach to enhance energy efficiency in near-/sub-threshold computing. The DPAD-EDaC strategy, with its deep path activity evaluation and adaptable prediction windows, effectively addresses the energy challenges in digital circuits. Its integration into a 32-bit microprocessor system demonstrates its robustness in achieving negative-margin operation with minimal overhead. The strategy’s adaptability, accentuated by clock gating, ensures effective error mitigation across voltage ranges. The significant energy savings at the MEP affirm its effectiveness in reducing consumption without compromising performance. Our study offers a valuable avenue for optimizing energy in near-/sub-threshold computing scenarios.

## Figures and Tables

**Figure 1 sensors-23-07498-f001:**
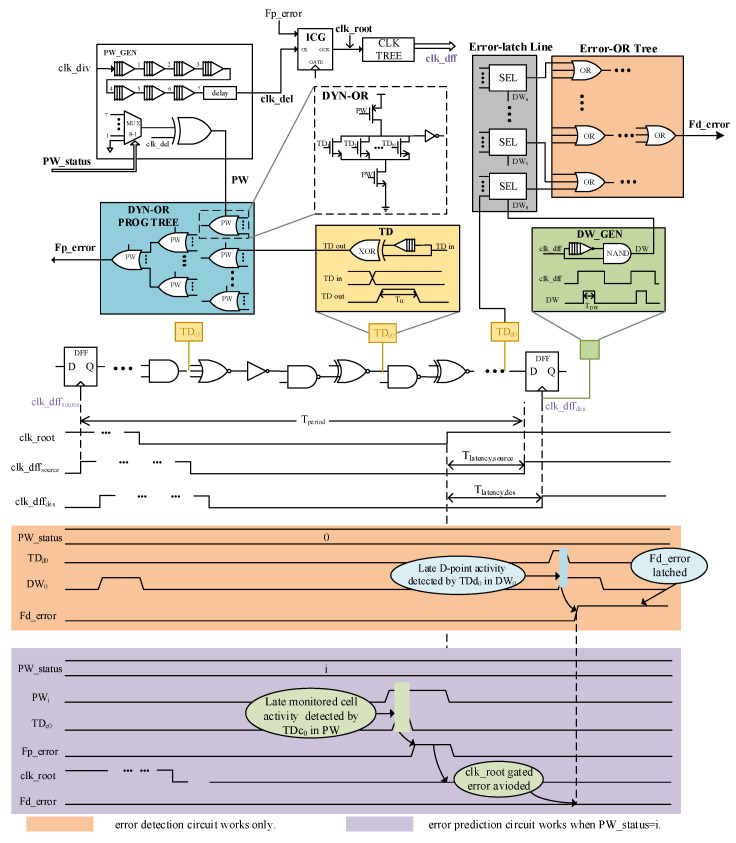
Detailed description of the specific implementation of the DPAD–EDaC strategy, where TDd0 and TDc0 identify late-arriving data changes that generate high prediction and detection error signals, respectively.

**Figure 2 sensors-23-07498-f002:**
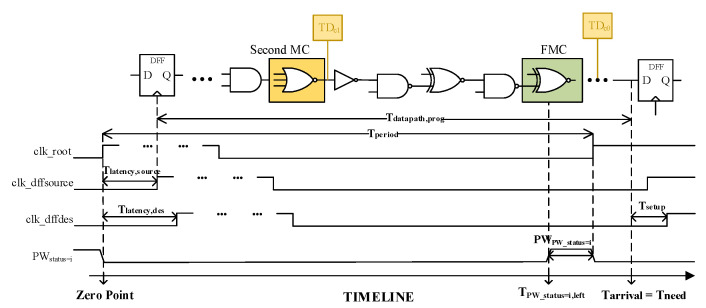
TIMELINE with clk_root as the origin. $T_{latency,source}$ represents the clock tree latency of the source endpoint. $T_{period}$ represents the clock period. $T_{datapath,prog}$ represents the propagation time of the critical path. $T_{setup}$ represents the setup time needed for proper operation. ${PW}_{PW\_status=i}$ represents the corresponding PW width.

**Figure 3 sensors-23-07498-f003:**
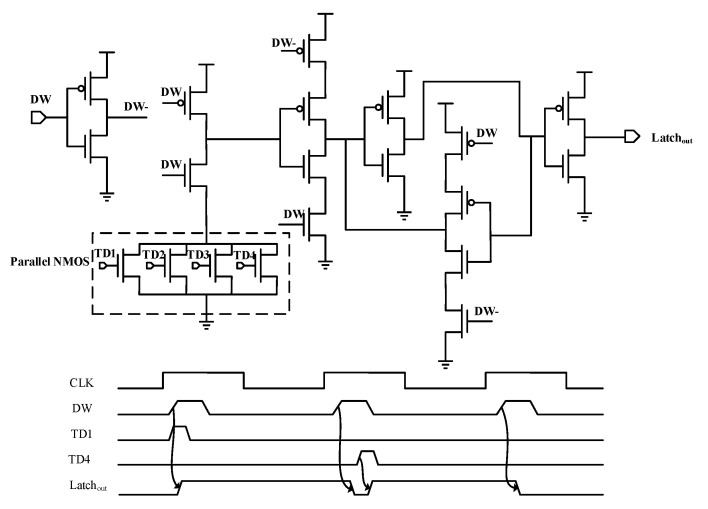
Schematic and timing diagram of the SEL cell. The fan-in of the cell changes with the number of NMOS transistors in the parrallel network.

**Figure 4 sensors-23-07498-f004:**
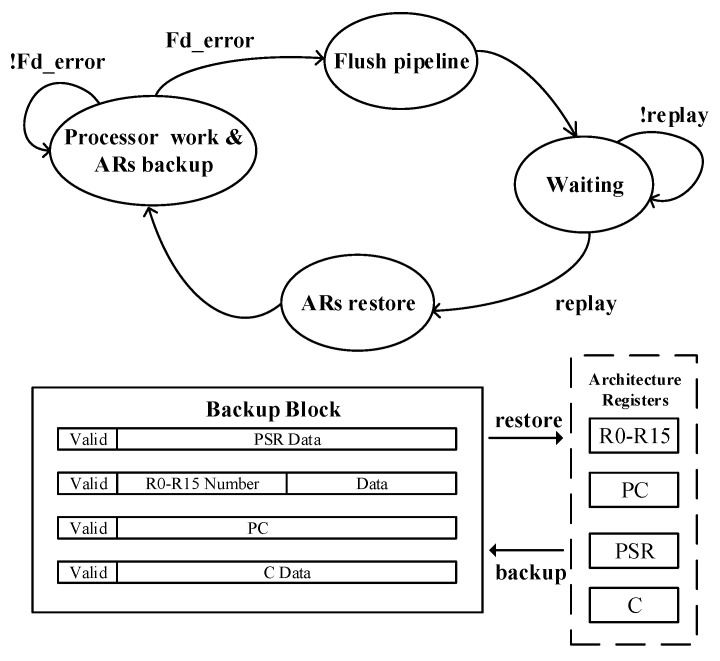
Illustration of the presented ECC. The valid bit in each row represents whether change in the corresponding AR has occured.

**Figure 5 sensors-23-07498-f005:**
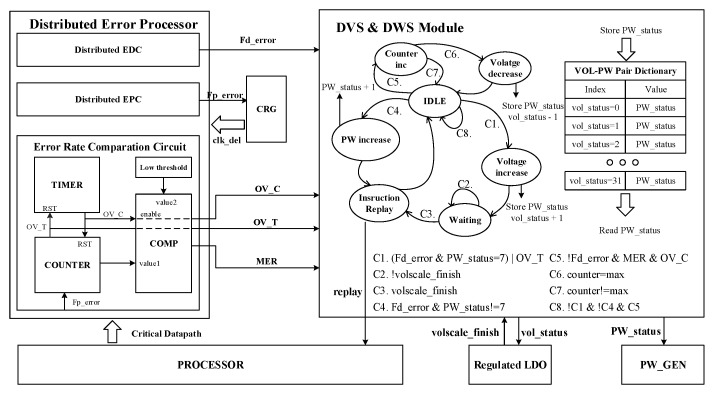
Detailed description of the DVS and DWS modules and the distributed error processor. The DEP obtains data activities from critical paths, and then generates prediction/detection error signals and prediction error rate for DVS and DWS module operation.

**Figure 6 sensors-23-07498-f006:**
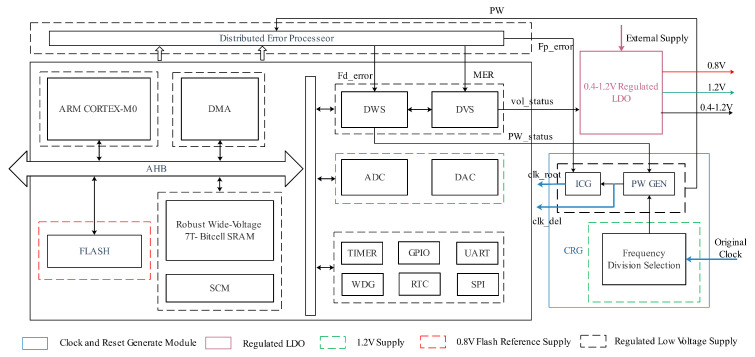
The 32-bit microprocessor system with DPAD–EDaC overview. The three colored dashed boxes represent the three voltage domains.

**Figure 7 sensors-23-07498-f007:**
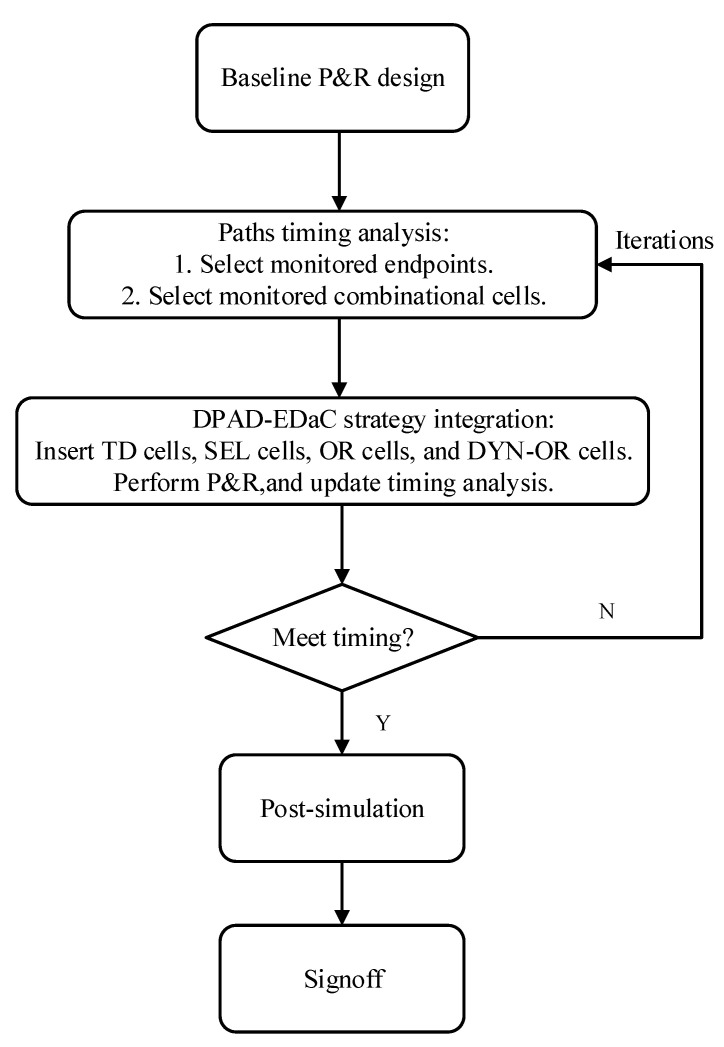
Automated flow for DPAD–EDaC integration. Baseline P&R design is the same as the conventional design.

**Figure 8 sensors-23-07498-f008:**
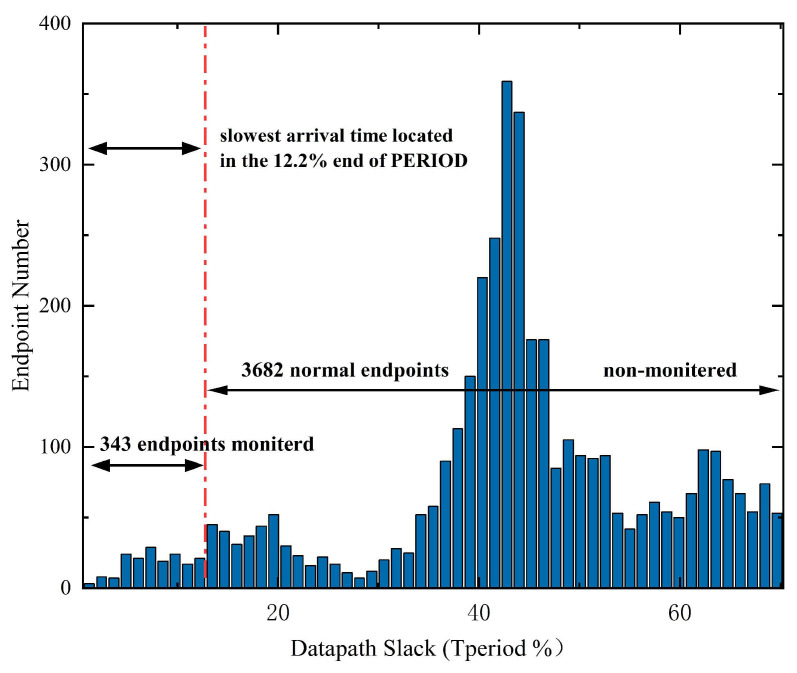
Histogram of the path with the smallest timing slack at each endpoint.

**Figure 9 sensors-23-07498-f009:**
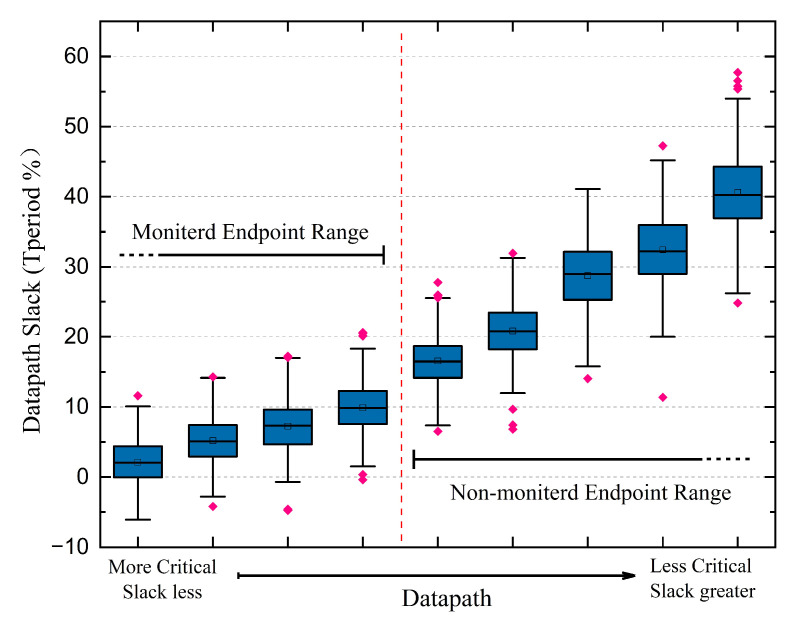
Boxplot of the slack distribution of a subset of paths, obtained from 1000 MC simulations at 0.5 V, TT corner.

**Figure 10 sensors-23-07498-f010:**
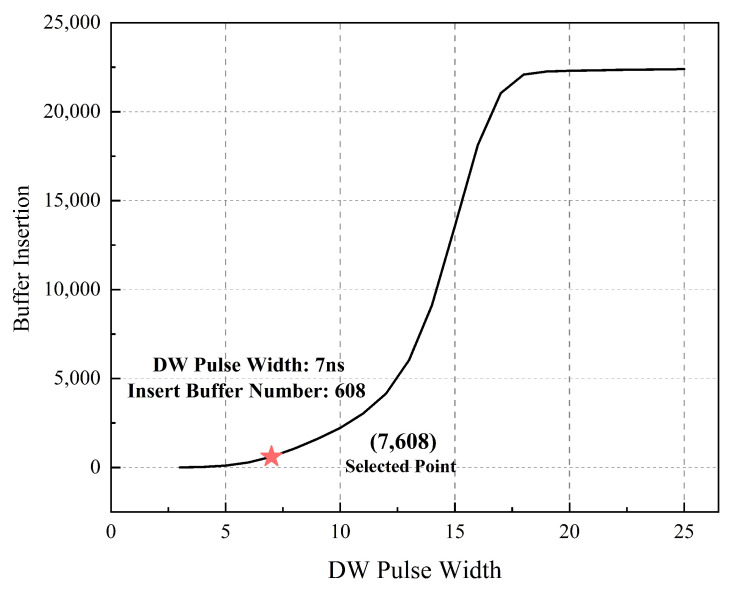
Hold buffers’ insertion number versus DW width.

**Figure 11 sensors-23-07498-f011:**
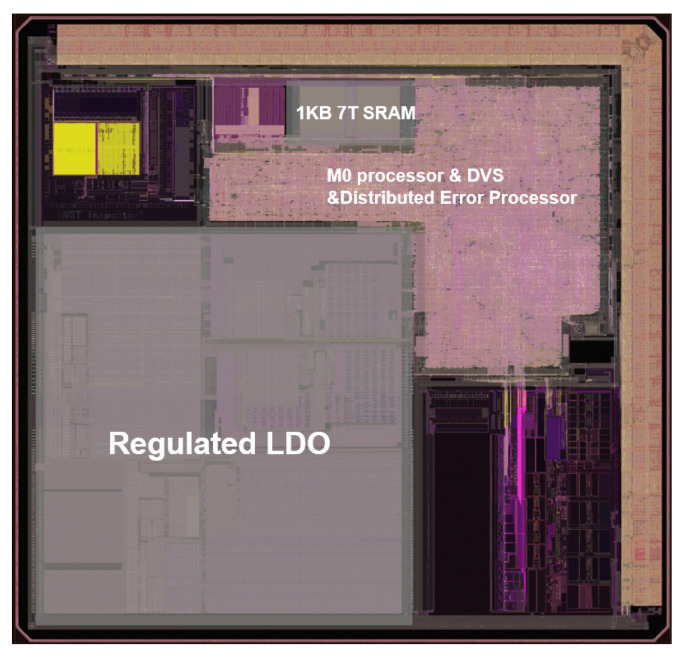
Chip micrograph.

**Figure 12 sensors-23-07498-f012:**
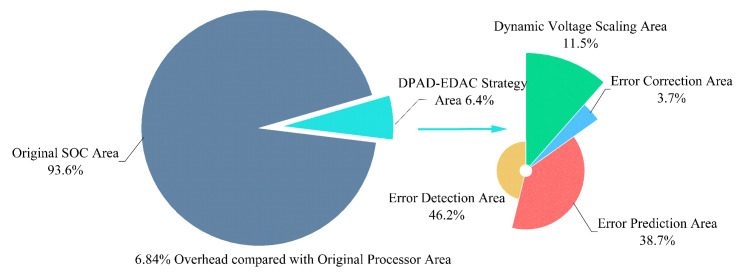
The components of DPAD–EDaC area overhead. The EDC accounts for 46.2% of DPAD–EDaC’s area overhead, which is the maximum.

**Figure 13 sensors-23-07498-f013:**
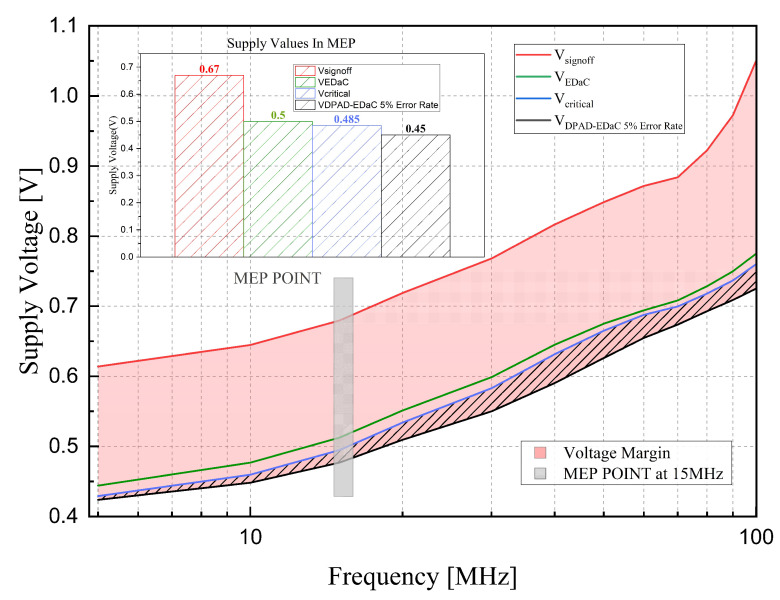
Voltage scaling and margin over frequency at four operation conditions. The voltage values in MEP are specially marked.

**Figure 14 sensors-23-07498-f014:**
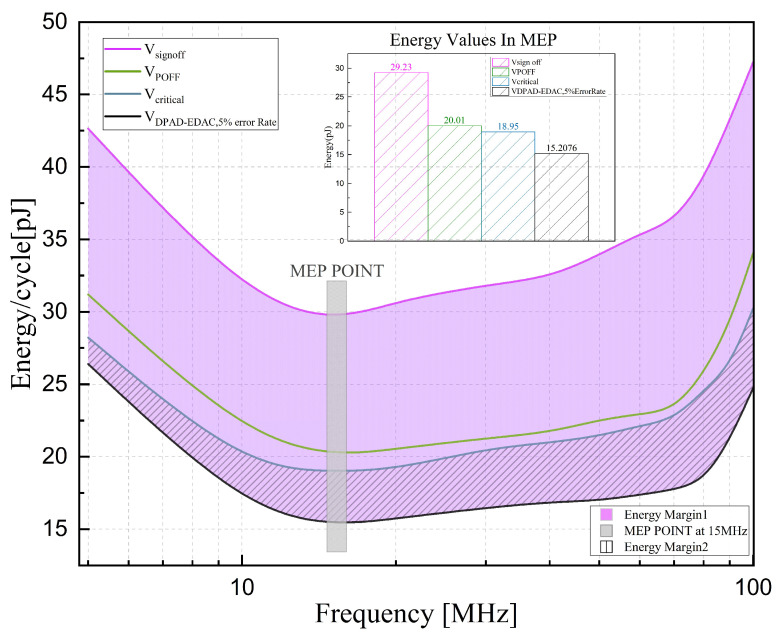
Energy scaling and margin over frequency at four operation conditions. The energy values in MEP are specially marked.

**Table 1 sensors-23-07498-t001:** Summary and comparison with existing near-threshold-capable EDaC systems.

	TVLSI’17 [31]	JSSC’18 [18]	JSSC’17 [23]	JSSC’19 [20]	JSSC’22 [25]	This Work
Method	EDFF	EDFF	Half path EP	EDL	CD TD	CD TD
DW(%TCLK)	-	5%	50%	50%	20%	7%
Correction	Replay	Borrowing	Predictive clock gating	Borrowing + Replay	Clock gating/Clock stretching	Clock gating + replay
Low-voltage failures ^1^	None	None	Failures	None	Failures	None
Area overhead	8.70%	7%	3.10%	4.17%	4.90%	6.84%
Technology	40 nm	40 nm	40 nm	28 nm	28 nm	55 nm
Gate count	145 K	M0/12 K	5 K	12 K	69 K	12 K
F-range (MHz)	27.4–286	5–30	40–750	18–68	1–200	1–100
V-range	0.6 V–1 V	0.29V–0.47V	0.44 V–1 V	0.4 V–0.9 V	0.25 V–0.65 V	0.4 V–1 V
Vdecrease (wrt Vsign) (wrt Ecritical) ^2^	23.10%	42%	18%	40%	22%	29.10%
Esave (wrt Esign)	44%	75%	50%	61%	33%	47.97%
Emargin remained (wrt Ecritical) ^3^	-	37%	-	-	12%	−19.75%

^1^ There are significant failures in strategy at both voltage and higher values. ^2^ The degree of voltage reduction attained by each method at the minimum energy point is compared to the supply voltage determined by conventional signoff criteria. ^3^ The conserved energy margin achieved by each method is assessed relative to the zero-margin reference point. Negative values denote further reductions in energy consumption.

## Data Availability

Not appliable.

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
