# Peer review of "Negative Design Margin Realization through Deep Path Activity Detection Combined with Dynamic Voltage Scaling in a 55 nm Near-Threshold 32-Bit Microcontroller"

_sensors, 2023, doi:10.3390/s23177498_

Round 1

Reviewer 1 Report

The authors present a timing error detection and correction (EDaC) strategy to reduce energy consumption. The work illustrates the method to improve performance, however, lacks clarity in terms of the flow of the paper, sentences used, and comparative analysis. Please work on the following suggestions:

Grammatical errors and spelling mistakes.

The title needs to be justified clearly in the Introduction. Please include a discussion on “negative margin”, and dynamic voltage scaling. While you indicate 15 reference papers that used different EDaC, the section should clearly highlight how it is different from other strategies as illustrated in Table 1.

Table I demonstrates a nice comparative study but requires results to be explained for a better understanding and to highlight the novelty of the work.

Reference should be cited in various places in the manuscript like the authors mention What’s more, 19.75% energy saving is obtained with respect to the zero-margined critical operation, which is a maximum of -12% in other works meaning that at least 12% energy loss. What are the other works you are referencing to?

This shows that the energy consumption is further compressed, and the negative margin is achieved. The table shows a positive margin. Either change the definition or clearly state what is meant by “negative”?

It needs to be improved in terms of spell check, grammar, and use of few adjective like "pessimism" which doesn't sound technical.

Reviewer 2 Report

This paper presents a timing error detection and correction strategy for the near-/sub-threshold operation to reduce energy consumption. The proposed strategy is implemented in a near-threshold 32-bit ARM Cortex microprocessor system and infers only 6.8% area overhead. Experimental results show that the system achieves a minimum energy point (MEP) of 15.2PJ/cycle at 15MHz and 0.475V.

The following comments can be taken into consideration to improve the quality of the paper:

1-   Obsolete and out-of-date References should be replaced with state-of-the-art and more up-to-date references related to the topic.

2-   The conclusion should be modified to be slightly different from the Abstract.

3-   A list of abbreviations should be included at the end of the paper.

4-   Limitations of the proposed method should be mentioned clearly. Also, is it applicable to all types of Microcontrollers and DSPs?

Reviewer 3 Report

The submitted article presents a novel approach to address the energy efficiency challenges in near-threshold computing (NTC). NTC offers energy savings but introduces path delay variations due to process, voltage, temperature variations, and aging. This work proposes an error detection and correction (EDaC) strategy that combines high error-aware capability with low area overhead and energy consumption. The contributions include a dynamic adjustment of the prediction window based on error detection results, a low-overhead error prediction circuit design, and the use of dynamic voltage scaling (DVS) to achieve energy savings while maintaining system throughput. The approach is implemented in a 32-bit ARM Cortex M0 microprocessor system and achieves significant energy reductions compared to both signoff margins and zero-margin operations. The article outlines the strategy's benefits and its practical implementation, supported by experimental results.

1.      Introduction Clarity: The introduction provides a comprehensive overview of the research context and objectives. However, consider simplifying some of the complex sentences to enhance clarity. Break down lengthy sentences into smaller ones to make the content more accessible to readers.

2.      Clarity of Figures: In several instances, figures like Figure 1, 2, 5, 6, 12, 13, and 14 appear to have unclear elements. To improve the visual clarity, consider enhancing the resolution or providing detailed captions that guide readers through the information presented.

3.      Avoid Unexplained Abbreviations: In the abstract and throughout the article, it's important to avoid using abbreviations without prior definition. For example, "EDaC" is not defined in the abstract. Ensure that all abbreviations are clearly introduced and explained upon first usage.

Rest of the paper looks good. I really like the author's comparing their results with other published work. 

The overall quality of English language in the article is good, but there are some areas where improvements could be made to enhance clarity and readability. The sentences are generally well-structured, but a few sentences appear lengthy and could be broken down for better comprehension. Additionally, in some instances, the use of complex vocabulary and sentence structures might hinder the clarity of the content for readers who are not experts in the field.

There are a few instances where incorrect tenses or inconsistencies in tense usage are present. Careful proofreading is needed to ensure that all sentences maintain a consistent tense throughout the article.
